# Extracellular Vesicle-Based SARS-CoV-2 Vaccine

**DOI:** 10.3390/vaccines11030539

**Published:** 2023-02-24

**Authors:** Yasunari Matsuzaka, Ryu Yashiro

**Affiliations:** 1Division of Molecular and Medical Genetics, The Institute of Medical Science, Center for Gene and Cell Therapy, University of Tokyo, Minato-ku, Tokyo 108-8639, Japan; 2Administrative Section of Radiation Protection, National Institute of Neuroscience, National Center of Neurology and Psychiatry, Kodaira, Tokyo 187-8551, Japan; 3Department of Infectious Diseases, Kyorin University School of Medicine, 6-20-2 Shinkawa, Mitaka-shi, Tokyo 181-8611, Japan

**Keywords:** drug delivery, exosomes, extracellular vesicles, lipid nanoparticle, SARS-CoV-2, viral vector

## Abstract

Messenger ribonucleic acid (RNA) vaccines are mainly used as SARS-CoV-2 vaccines. Despite several issues concerning storage, stability, effective period, and side effects, viral vector vaccines are widely used for the prevention and treatment of various diseases. Recently, viral vector-encapsulated extracellular vesicles (EVs) have been suggested as useful tools, owing to their safety and ability to escape from neutral antibodies. Herein, we summarize the possible cellular mechanisms underlying EV-based SARS-CoV-2 vaccines.

## 1. Introduction

Coronaviruses, enveloped viruses with a single, positive-strand RNA, are common in animals, including humans [1]. These viruses are spherical, with a diameter of about 100 nm, and have distinctive crown-like projections or spikes on their surfaces, hence the name “corona” (Latin for crown) [2]. Of the coronaviruses known to be infectious and pathogenic to humans, SARS-CoV-2, the causative virus of coronavirus disease 2019 (COVID-19), utilizes S-glycoprotein [3]. S-glycoprotein contains two functional domains: an S1 receptor-binding domain (RBD) and a second S2 domain mediating the fusion of viral and host cell membranes to allow viral entry into host cells [4]. The S protein of SARS-CoV-2 first binds to the angiotensin-converting enzyme 2 (ACE2) receptor on the surface of the host cell via the S1 RBD. Then, the S1 domain is shed from the viral surface, allowing the S2 domain to fuse with the host cell membrane [5]. This fusion depends on the activation of the S protein by cleavage at two sites, S1/S2 and S2′, mediated by the proteases furin and transmembrane protease serine 2 (TMPRSS2) [6,7]. Furin cleavage at the S1/S2 sites leads to conformational changes in the S protein, exposing the RBD and S2 domains, and cleavage of the S protein of SARS-CoV-2 by TMPRSS2 enables the fusion of the capsid with the host cell, allowing the virus to enter the cell. Exposure of the RBD of the S1 protein subunit produces an unstable subunit conformation. As a result, this subunit undergoes a conformational rearrangement between two states, which are temporarily hidden or release RBD upon binding. Within the trimeric S protein, only one of the three RBDs can bind to the human ACE2 host cell receptor [4].

The infectivity of SARS-CoV-2 compared to SARS-CoV can be explained by the large number of receptors that, in addition to binding to ACE2, allow SARS-CoV-2 to bind to other cell surface molecules. For example, both neuropilins 1, expressed in neurons, allow viruses to enter the nervous system and bind to truncated forms of the S protein of SARS-CoV-2 to mediate host cell entry [8]. In addition to the known ACE2 receptors, binding to cell surface neuropilin receptors enhances SARS-CoV-2 infection [9]. Furthermore, the S protein of SARS-CoV-2 binds to CD147 on the cell surface, the expression of which is increased by high blood glucose levels, and subsequently enters cells, which is indicative of the poor prognosis of diabetics with COVID-19 [10]. In addition, the frequent targets of SARS-CoV-2 in humans are nerve, vascular endothelial, and epithelial cells in the respiratory and gastrointestinal tracts. However, ACE2 is expressed only at low levels in the brain. SARS-CoV-2 can also bind to cell surface sialic acid glycoproteins and gangliosides, including neurons, which are highly expressed on the surface of all cell types targeted by SARS-CoV-2 [11,12]. These results suggest that sialic acid levels play an important role in SARS-CoV-2 cell entry.

Different types of currently used vaccines, such as inactivated vaccines, recombinant protein vaccines, and peptide vaccines, utilize a mechanism that allows the human body to be immunized by administering a portion of the viral protein [13]. In messenger RNA (mRNA) and viral vector vaccines, part of the genetic information, which is the basis for producing viral proteins, is injected [14]. Based on this information, some viral proteins are produced in the human body in addition to the antibodies against them, thereby establishing immunity. In the mRNA vaccine against SARS-CoV-2, the mRNA, which is the blueprint for the spike protein, is encapsulated in a lipid membrane. When this vaccine is inoculated and mRNA is taken up into human cells, a spike protein is produced within the cell based on this mRNA. By inducing neutralizing antibody production and cell-mediated immune responses against the spike protein, it becomes possible to prevent infectious diseases caused by SARS-CoV-2. Moreover, in the viral vector vaccine, the gene encoding the amino acid sequence of the SARS-CoV-2 spike protein is incorporated into the viral vector. When this vaccine is inoculated and the gene is incorporated into human cells, a spike protein is produced in the cells based on this gene. Similarly to mRNA vaccines, it is possible to prevent SARS-CoV-2 infection by inducing neutralizing antibody production and cell-mediated immunity against the spike protein. However, several issues, including the efficacy and side effects of these vaccines, remain unaddressed.

In addition, extracellular vesicles (EVs), as communication tools derived from living organisms, are attracting attention in various fields, including medicine, for the treatment and diagnosis of diseases [15]. In particular, the possibility of manufacturing safe and effective vaccines by encapsulating nucleic acids and viruses has been suggested, and expectations for SARS-CoV-2 vaccines are increasing. In this review, we summarize the advancements in and molecular mechanisms of SARS-CoV-2 vaccination with EV encapsulation of target molecules.

## 2. Classification of Viruses

Viruses are officially classified by the International Committee on Taxonomy of Viruses based on their molecular biological properties, including genomic composition and nucleotide sequence similarity [16]. Coronaviruses are classified within the family Coronaviridae of the order Nidoviridae and suborder Cornidoviridae. The family Coronaviridae is further divided into the subfamilies Retovirinae and Orthocoronavirus, and the latter includes four genera: alpha (α), beta (β), gamma (γ), and delta (δ) (Figure 1) [17]. MERS-CoV, a bat coronavirus that acquired pathogenicity in humans, was first isolated in 2012 from a patient in the UK with severe pneumonia who had a history of staying for long periods in the Middle East [18]. MERS-CoV is the causative virus of Middle East Respiratory Syndrome (MERS), with a fatality rate of 35%. MERS is transmitted to humans via dromedary camels, the natural host; however, human-to-human transmission can also occur, and an outbreak occurred in South Korea in 2015.

SARS-CoV-2 is the causative virus of coronavirus disease 19 (COVID-19), and SARS-CoV-1 is the causative virus of severe acute respiratory syndrome (SARS), which was first reported in 2002–2003 and spread throughout the world [18,19]. Bats are natural hosts, and humans are infected through contact with infected bats or other animals infected by bats. Thus, SARS, MERS, and the new COVID-19 coronavirus are all zoonotic viruses transmitted from mammals, the natural hosts, to humans. Both SARS-CoV-1 and SARS-CoV-2 are classified into the β-coronavirus subgenus *Sarbecovirus* and use ACE2 on the surface of human cells to infect humans. Five types of viruses are currently categorized as coronaviruses which infect humans. Four of the identified human pathogenic coronaviruses (HcoV-229E, HcoV-NL63, HcoV-OC43, and HcoV-HKU1) are cold coronaviruses [20]. Apart from MERS-CoV, whose receptor is dipeptidyl peptidase, HcoV-229E, HcoV-NL63, HcoV-OC43, and HcoV-HKU1 are human coronaviruses (HCoVs) that routinely infect humans. Of these, HcoV-229E and HcoV-NL63 are members of the *α-coronavirus* genus. HcoV-OC43, HcoV-HKU1, and MERS-CoV belong to the *β-coronavirus* genus, which is divided into four lineages (A, B, C, D), with HcoV-OC43, HcoV-HKU1, SARS-CoV-1, and SARS-CoV-2 belonging to the B lineage. Viruses isolated from wild bats in China in 2013 (bat/Yunnan/RaTG13/2013) and Malayan pangolins in 2019 are closely related to SARS-CoV-2 and belong to the B lineage [21,22,23,24]. Phylogenetic analysis suggests that all of the human coronaviruses mentioned above originated from wild animals, such as bats and rodents. Coronaviruses, which originally existed in natural hosts such as bats and rodents, first infected intermediate hosts before eventually infecting humans and causing diseases. As for SARS-CoV-2, sequences of coronaviruses closely related to this virus have been found in bats; therefore, it is highly likely that bats are also the natural hosts of SARS-CoV-2. In addition, since a coronavirus closely related to SARS-CoV-2 has been detected in Malayan pangolins, it is theorized that Malayan pangolins are an intermediate host; however, this remains to be substantiated.

## 3. Structural Characterization of SARS-CoV-2

### 3.1. Inner and Outer Structure of SARS-CoV-2

SARS-CoV-2 has a spherical shape of approximately 100 nm, consisting of a lipid bilayer membrane, called an envelope, surrounding the viral sphere, with spike (S), envelope (E), and membrane (M) proteins that pierce this membrane and bind to receptors [25,26]. The spike protein is the outermost structure of the viral particle. When the human immune system produces antibodies, it recognizes this spike protein and produces antibodies that match its shape. Inside the sphere is an RNA genome of approximately 30 kb, the largest known RNA viral genome, bound to a protein called nucleocapsid, which consists of the N protein, the most abundant protein in viral particles [27,28]. The outer nucleocapsid is enveloped and consists of lipid membranes, which are easily destroyed by washing with soap and detergent or disinfecting with alcohol, rendering the virus infective. Genes encoding nonstructural proteins, such as enzymes essential for viral replication, RNA polymerase, and protease, are present at the 5′-end, and genes encoding structural proteins, such as S, E, M, and N, are present at the 3′-end. Accessory genes also exist in regions where structural protein genes exist, and they encode proteins that contribute to the optimization of viral replication [29].

### 3.2. SARS-CoV-2 Receptor

The homology of SARS-CoV-2 genomic RNA and viral proteins with SARS-CoV-1 is 79.0% for genomic RNA, 76.2% for S protein, 94.7% for E protein, 90.1% for M protein, and 90.1% for N protein. Similar to SARS-CoV-1, the S protein of SARS-CoV-2 enters host cells by binding to human ACE2 through complex formation by binding the receptor-binding domain (RBD) of the S protein and ACE2 [30,31]. However, the S protein of SARS-CoV-2 does not recognize the MERS-CoV receptor DPP4 or the HcoV-299E receptor APN. The RBD of the SARS-CoV-1 S protein is composed of a core structure and a receptor-binding motif (RBM) that directly binds to the ACE2 surface [32]. Furthermore, six amino acids, Y442, L472, N479, D480, T487, and Y491, of the RBD of SARS-CoV-1 are important for binding to ACE2 and involved in determining the host range of SARS-related coronaviruses. In SARS-CoV-2, the corresponding six amino acids are L455, F486, Q493, S494, N501, and Y505. Analysis of the binding affinity between the RBM of SARS-CoV-2 and ACE2 in various animal species, including humans, revealed that the RBM of SARS-CoV-2 had a high binding affinity with ACE2 in humans, muskrats, pigs, ferrets, cats, orangutans, green monkeys (*Chlorocebus sabaeus*), and bats (Rhododendronidae) and a low binding affinity with mouse and rat ACE2. One of the major differences between SARS-CoV-2 and SARS-CoV-1 is the characteristic sequence of consecutive basic amino acids (RRAR) at the S1/S2 cleavage site of the SARS-CoV-2 S protein, called the “furin cleavage site”, which is efficiently cleaved by furin and other proteases (Figure 2) [6,33]. This furin cleavage site is absent in SARS-CoV-1, but is present in the S proteins of MERS-CoV and HcoV-OC43. During the viral replication cycle, the S protein is cleaved into S1 and S2; however, the location and timing of cleavage differ depending on the type of coronavirus. In infected cells, the S protein is synthesized and cleaved by host proteases, or the S protein binds to the receptor and is then cleaved by host proteases when the virus enters the target cell. Since SARS-CoV-1 uses the latter mechanism, the S protein is present on the surface of the viral particle in an uncleaved state. When the virus invades the cell, the S protein is cleaved by host proteases, trypsin, elastase, cathepsin, and TMPRSS2. In contrast, in the case of SARS-CoV-2, cleavage occurs within the cell after S protein synthesis via the first mechanism. The furin cleavage site of the S protein is necessary for SARS-CoV-2 to efficiently infect the human respiratory tract, and the activation of the S protein by TMPRSS2 is also important [34].

### 3.3. Replication Mechanism of SARS-CoV-2

SARS-CoV-2 is obligately intracellular and cannot multiply in the virion state. It needs a parasite to enter a host cell, and then uses the functions of that cell to multiply itself with the following steps. (i) Absorption: Viruses must first bind to receptors on animal cell surfaces [35,36,37]. In SARS-CoV-2, this life cycle begins when the surface-protruding S protein binds to the receptor of ACE2, an enzyme protein on the cell surface. The receptor that recognizes and binds to the virus is determined by the virus. For example, in the influenza virus, the structure of sialic acid sugar chains on the surface of upper and lower respiratory tract cells functions as a viral receptor [38], and in HIV, a protein called CD4 on immune cells functions as a viral receptor [39]. ACE2 is involved in the breakdown of a blood pressure-regulating autocoid called angiotensin II; however, the infection does not cause sudden changes in blood pressure. The sequence cleaved by a human-produced protease, furin, does not exist in SARS-CoV-1. However, it does exist in the S protein of SARS-CoV-2, and owing to the cleavage by furin, the S protein binds ACE2 strongly and spreads infection. (ii) Entry: After the S protein binds to the ACE2 receptor, an enzyme called TMPRSS2, a type II transmembrane serine protease, on the cell surface cleaves part of the S protein, altering its binding [7,40,41]. This triggers the fusion of the viral envelope with the cell membrane, allowing it to enter the cell. (iii) Uncoating: Infectious viral particles are decomposed and eliminated, and genomic RNA is exposed and released into the cytoplasm of the host cell. In the case of SARS-CoV-2, it occurs almost simultaneously with step (ii). (iv) Synthesis of the materials: This process involves the production of various nucleic acids, proteins, and other components of viral particles from the viral genome. Because the genome of SARS-CoV-2 is RNA-based, it consists of the following steps. (iv-a) Synthesis of RNA-dependent RNA polymerase (RdRp): SARS-CoV-2, which has (+)-strand RNA in its genome, first synthesizes RdRp encoded in the viral genome using host ribosomes [42,43,44,45,46,47]. Unlike DNA-dependent RNA polymerases in the host nucleus, RdRp is an enzymatic protein that synthesizes new RNA while reading the RNA sequence of the viral genome. (iv-b) Replication of genomic RNA: RdRp in (iv-a) synthesizes (−)-strand RNA from the (+)-strand RNA genome and further synthesizes (+)-strand genomic RNA using the (−)-strand as a template. (iv-c) Transcription and translation: In addition to genomic RNA, several short subgenomic mRNAs functioning as mRNAs are synthesized and utilized for viral protein synthesis via human ribosomes. (iv-d) In most cells, including human cells, the information for one type of protein is on one mRNA, and the ribosome synthesizes the protein based on this information. However, in SARS-CoV-2, information for multiple proteins is carried simultaneously on each subgenomic mRNA, and the sequence information of one long protein is connected. The ribosome of the host cell interprets all this information and synthesizes a single polypeptide chain. This chain is cut into individual proteins by the action of a protease that cleaves a specific sequence. In addition, this protease inhibitor has been put into practical use as an anti-HIV and anti-HCV drug, and is expected to be effective against SARS-CoV-2 [48,49,50,51]. (v) Assembly/maturation and (vi) release of viral particles: Among the various parts that make up the viral particle, those that are incorporated into the envelope are synthesized on the rough endoplasmic reticulum and incorporated into the endoplasmic reticulum membrane. In addition, the capsid protein binds to the viral genome RNA to form a nucleocapsid, and the endoplasmic reticulum membrane containing the viral proteins is cut off to surround it, forming viral particles, which are released outside the cell by exocytosis via the Golgi apparatus. After going through steps (i) to (vi), the virus is amplified several hundred times in the cell, released outside the cell, and then adsorbed into the surrounding uninfected cells to repeat the infection process.

Viruses other than SARS-CoV-2 cannot grow alone and can only grow inside host cells. Unlike bacteria, they cannot multiply via cell division. Establishing a method to inactivate an active virus is difficult because the active virus mostly uses the human host system. If it is destroyed, it will also destroy the human system, and there are not many effective means for treating viruses alone. Selective toxicity refers to the property of affecting only the target; however, antiviral drugs have side effects, making it difficult to develop drugs with high selective toxicity. Despite this, the viral life cycles described by steps (i) to (vi) also contain several promising target proteins as potential sites of action for highly selective and toxic antiviral drugs. For example, RdRp is not present in the human system and is, therefore, a promising target. Fusan (nafamostat), already in use as an inhibitor of TMPRSS2, is expected to be an effective SARS-CoV-2 drug [52]. Furthermore, protease inhibitors have been put into practical use against HIV and HCV, and further progress in research is expected.

### 3.4. Viral Characterization Using Animal Models

Dozens of coronaviruses infect mammals other than humans and birds and are classified in either genus. However, so far, only α or β coronaviruses have been isolated and identified as pathogenic to humans. Coronaviruses have a wide host range, infecting not only humans and wild animals but also livestock, pets, and experimental animals, causing various diseases. SARS-CoV-2 has been detected in two pet dogs and one cat in Hong Kong, as well as in two pet cats and various feline animals, such as tigers and lions, in zoos across New York. From the analysis of susceptibility to SARS-CoV-2 in animal species close to humans, such as livestock and pets, the virus proliferated in the respiratory organs of ferrets and cats, but not in dogs, pigs, chickens, and ducks. SARS-CoV-2 causes droplet transmission, especially in ferrets [53,54]. Furthermore, SARS-CoV-2 is highly proliferative in the respiratory tracts of cats and can be easily transmitted among cats by contact; however, cats infected with this virus do not show obvious symptoms [55]. In the hamster model of the SARS-CoV-2 infection system, the virus proliferated well in the respiratory tract and caused lesions in the lungs, similar to those of human patients with COVID-19. In addition, non-infected hamsters treated with convalescent sera from infected individuals inhibited SARS-CoV-2 proliferation [56].

## 4. Virulence of SARS-CoV-2

SARS-CoV-2, similar to other SARS viruses, is mainly transmitted by droplet and contact infections [57]. The basic reproduction number, indicating the transmissibility of the virus itself, is approximately two to three, and no examples of double-digit estimations have been described, such as for the measles virus, which is airborne. Accordingly, airborne transmission does not occur; however, the infection may occur through aerosols, which are smaller than droplets. SARS-CoV-2 shows a wide range of disease symptoms, from asymptomatic cases to cold and pneumonia, with aggravation of severe cases leading to death [58]. Respiratory symptoms and oxygen saturation are used as indicators for severity classification; however, the following mechanisms are possibly involved in SARS-CoV-2 infection, causing dyspnea [59,60,61].

Respiratory failure due to pneumonia exacerbationAcute respiratory distress syndrome due to cytokine stormsPulmonary blood flow disorders due to thrombosis/embolisms

Of these, the first and second often occur in other viral pneumonia, and the third may be closely related to the characteristic nature of SARS-CoV-2.

D-dimer levels increase with COVID-19 infection; thus, they represent a useful marker of aggravation [62,63,64,65]. D-dimer is a fibrin breakdown product formed by blood clotting. In addition, autopsy findings of patients who had died of COVID-19 showed thrombus formation in the alveoli and deep veins. Therefore, COVID-19 may cause increased intravascular blood coagulation. This suggests that a cytokine storm causes dysfunction of vascular endothelial cells, resulting in disseminated intravascular coagulation (DIC) [64,65,66]. However, there are some exceptions, such as cases with elevated D-dimer levels that are not seen in DIC, and with elevated D-dimer levels, even in mild cases without a cytokine storm. Combining these clinical findings and the fact that ACE2 is a novel SARS-CoV-2 infection receptor expressed in vascular endothelial cells, there are cases in which a virus that has directly entered the blood infects vascular endothelial cells and causes vascular injury, thereby inducing thrombus formation [67,68]. Furthermore, findings of SARS-CoV-2 infection in vascular endothelial cells and associated vasculitis have been reported. Therefore, ACE2, by acting as a receptor, possibly causes aggravation through thrombus formation. Based on these findings, rapidly progressing dyspnea causes pulmonary thromboembolism, complications such as cerebral infarction and myocardial infarction, arterial thrombosis, and frostbite-like skin lesions caused by ischemia due to thrombosis in peripheral blood vessels [69,70,71,72,73]. In addition, if irreversible changes occur in various tissues due to blood flow disturbances, sequelae remain even after the virus disappears. Considering this, SARS-CoV-2 can not only cause pneumonia, but can also cause various systemic diseases. This does not necessarily mean that the virus is unmanageable. Rather, it suggests the possibility of treating it using appropriate measures against blood clots. Indeed, the combination of D-dimer testing and anticoagulant therapy has been successful in preventing and treating severe diseases.

Furthermore, other coronaviruses also use ACE2 as a receptor. For example, the receptor for the SARS coronavirus, closely related to the novel SARS-CoV-2 coronavirus, is also ACE2. SARS infected approximately 8000 people worldwide, far fewer than SARS-CoV-2, with deaths recorded about eight months after initial reports. The number of deaths was less than 800, and there were few reports on pathological autopsies; however, cases of thrombus formation and vasculitis in various organs were reported. In contrast, coronavirus NL63, which also uses ACE2 as a receptor, mainly causes upper respiratory tract infections and reportedly does not cause severe pneumonia or thrombosis/embolisms [74,75,76]. The S protein of NL63 has a weaker binding ability to ACE2 than the SARS coronavirus, and it may have difficulty infecting the lower respiratory tract and moving into the blood.

## 5. Vaccination of SARS-CoV-2

Humans are equipped with two types of immune systems to defend against foreign bodies, such as bacteria and viruses: innate immunity and acquired immunity [77,78,79,80,81,82,83,84,85,86,87,88]. Innate immunity is a mechanism whereby the body defends itself by recognizing the general characteristics of invading viruses and bacteria, which are then phagocytized by phagocytizing cells, such as white blood cells, neutrophils, and macrophages. Acquired immunity is a mechanism that eliminates foreign substances more efficiently than innate immunity. It makes use of more specialized immune cells, such as B cells, that eliminate pathogens using antibodies, helper T cells that help B cells to make antibodies, and killer T cells that eliminate pathogens by killing infected cells. Later, some of the B and T cells become memory cells that can remember past infections, thus ensuring a more efficient defense upon encountering the same pathogen a second time. The purpose of vaccination is to induce these memory cells, which are divided into following main types: antibody-producing cells, memory B cells, and memory T cells (helper and killer T cells) [89,90,91,92,93,94,95,96,97,98]. Antibody-producing cells are specialized cells that produce antibodies. The number of antibodies present in the body, proportional to the number of antibody-producing cells, plays a significant role in protecting the body from viruses. When vaccination is used to induce the production of a sufficient number of antibodies in advance, a high infection prevention effect can be obtained. However, the number of antibody-producing cells produced by the SARS-CoV-2 vaccine gradually decreases, and after more than half a year, the antibody concentration drops to approximately one-fourth of the peak level. Although antibody concentrations decline over time, some memory cells produce large amounts of new antibodies after infection; these are called memory B cells. In addition, another type of memory cell, the memory T cell, can either become a helper T cell that helps memory B cells to change into antibody-producing cells or a killer T cell that eliminates viruses from infected cells. Thus, multiple types of memory cells can support the defense of antibody-producing cells. Among these memory cells, some cells do not decrease in number, and even if the antibody concentration drops significantly, the onset prevention effect is maintained at a certain level or higher owing to the presence of these memory cells. Nonetheless, booster vaccination is recommended, as the protective effect diminishes over time. While many COVID-19 vaccines are being developed around the world, at present, the main vaccines are mRNA- and viral vector-based vaccines [99,100,101,102,103,104,105,106,107,108,109,110,111,112,113,114,115,116]. Vaccine development is progressing using various methods, including conventional inactivated and recombinant protein vaccines.

### Mechanism of Action of Live and Inactivated Vaccines

Vaccines are methods of biological preparation containing attenuated or killed pathogens, and their components are administered to confer pathogen-specific immunity, effective only against targeted pathogens. Live vaccines that use attenuated pathogens are called inactivated vaccines, which use killed pathogens or their components [102,117,118]. In addition, vaccines using infectious viral vectors have been developed. The components of pathogens used in previous inactivated vaccines were mainly proteins and polysaccharides, whereas the COVID-19 vaccine (Pfizer and Moderna) used mRNA. Genetic vaccines, such as mRNA vaccines and viral vector vaccines, have the advantage of being put into practical use quickly, and are useful in pandemic scenarios requiring urgency [99,100,101,102,103,104,105,106,107,108,109,110,111,112,113,114,115,116]. mRNA vaccines using the S protein present on the surface of SARS-CoV-2 have been developed and have shown promising results. Generally, in viral infections, antibodies prevent the virus from entering the body. When a virus invades the body, the immune cells, specifically CD8-positive cytotoxic T cells (killer T cells), recognize some of the antigens presented by human leukocyte antigens on infected cells and destroy all infected cells to prevent the disease from spreading. This antigen fragment, called the epitope, is a specific peptide structural unit of the virus consisting of several amino acids. Therefore, it is important to identify the epitopes that prevent severe COVID-19. Since mRNA is easily destroyed by RNases in the human body, it must be encapsulated by wrapping it in lipid nanoparticles (LNPs) after modifying and optimizing its structure to facilitate its uptake into human cells to prevent degradation. mRNA vaccines are administered by intramuscular injection. Proteins are translated using mRNA as a template in immunocompetent cells, such as muscle and dendritic cells, and some of the produced proteins are presented to lymphocytes, causing an immune response. In addition, mRNA and LNP lipids act as adjuvants to stimulate innate immunity, resulting in immune induction. The mRNA vaccines manufactured by two pharmaceutical companies, Pfizer and Moderna, use the entire S protein gene. The S protein is produced in muscle cells and antigen-presenting cells, and specific antibodies against spike proteins are induced in vivo. For SARS-CoV-2 to enter human cells, it must bind to ACE2 on human cells. Vaccine-induced specific neutralizing antibodies against the S protein block SARS-CoV-2 entry into cells. In addition, humoral immunity by antibodies and cellular immunity by cytotoxic T lymphocytes is induced.

In addition, a viral vector vaccine incorporates a specific gene into an infectious virus, such as adenovirus or an adeno-associated virus, and is administered to the human body. It has already been applied in the treatment of congenital metabolic diseases and cancer. Moreover, it has been put into practical use as a vaccine for Ebola hemorrhagic fever. Similarly to mRNA vaccines, proteins are synthesized from genes in human cells, and an immune response occurs. Viruses as vectors and carriers themselves are not pathogenic; however, some can replicate and multiply within the human body, while others cannot. The AstraZeneca plc viral vector vaccine against SARS-CoV-2 uses chimpanzee adenovirus, while the Johnson & Johnson vaccine uses adenovirus, which cannot replicate in humans. The vector contains the entire gene encoding the SARS-CoV-2 S protein, which induces humoral and cellular immunity against the S protein. In addition, there are other vaccines, such as live vaccines and inactivated vaccines, which are preparations that weaken the pathogenicity and toxicity of live viruses as much as possible so that symptoms do not occur. In these types of vaccines, pathogenicity has been lost through the processes of heat treatment, phenol addition, formalin treatment, and ultraviolet irradiation for pathogens of cultured viruses. They have the advantages that the effects of live vaccines are likely to be obtained due to induction of immunity in a state close to natural infection, and that inactivated vaccines do not multiply in the body after vaccination and are highly safe. On the other hand, there are some issues; attenuated pathogens can multiply in the body, causing live vaccines to result in certain symptoms, such as fever and rash. They may also cause multiple doses and additives, called adjuvants, to be required due to their lower efficacy compared to line vaccines.

## 6. Drug Delivery System by Extracellular Vesicle

### 6.1. Biogenesis and Characterization of EVs

EVs, nano- to micro-sized particles surrounded by a lipid bilayer membrane secreted by almost all cells, play a role in transporting functional molecules with physiological activity from cell to cell [119,120,121,122,123]. EV is a general term for membrane vesicles secreted by cells, and there are various types of particles depending on the mechanism of formation, size, and molecular composition. EVs include exosomes, which are between 30 and 150 nm in diameter, derived from multivesicular endosomes. Exosomes are not formed directly from the cell membrane but are formed intracellularly and then secreted out of the cell. In addition, exosomes are formed by budding inside early endosomes from the cytoplasm, and ESCRT (endosomal sorting complex required for transport) and tetraspanins are involved in their formation [124]. In addition, endosomes containing many exosomes are called multivesicular bodies (MVB) because of their shape. Since the lipids that comprise the exosome membrane are rich in ceramide, sphingomyelin, and cholesterol, and are similar to lipid rafts, they are thought to form from lipid raft-like regions of the MVB. In fact, overexpression of nSMAse2/Smpd3 (neutral sphingomyelinase 2/sphingomyelin phosphodiesterase 3), a ceramide synthase, increases exosome secretion. Apoptotic vesicles, the largest heterogeneous population, are between 50 and 5000 nm in diameter and are generated from cell fragments during programmed cell death; they are produced by direct budding (shedding) from the cell membrane [124,125,126]. In addition, the existence of nanoparticles, called exomeres, with sizes of approximately 50 nm and without a membrane structure has also been reported [121,127]. EVs have membrane proteins and glycolipids on their surface, as well as various proteins and nucleic acids, such as DNA, mRNA, and miRNA. At the time of their discovery, EVs were thought to represent waste disposal mechanisms in cells, and it was found that intercellular communication is carried out by carrying the contents from one cell to another, which plays a critical role in various phenomena of life, such as immune response and signal transduction. In addition, many physiologically active substances have therapeutic effects on various diseases, and EVs are attracting attention as new biopharmaceuticals. In particular, EVs derived from mesenchymal stem cells harbor miRNAs and proteins associated with physiological and pathological processes, such as epigenetic regulation, immune regulation, and tumor formation and progression [128,129,130,131,132,133,134]. Furthermore, EVs have properties suitable for drug delivery compared to synthetic nanoparticles and are expected to be used as novel drug delivery system (DDS) nanocarriers. When the patient’s own EVs are used as carriers, their membrane composition is derived from autologous cells; therefore, they are less immunogenic and more biocompatible than synthetic nanoparticles. In addition, small-sized EVs can prevent monocytes from phagocytosis, extravasate from tumors into blood vessels, accumulate passively in tumor tissue, and penetrate in vivo barriers, such as the blood–brain barrier [135,136]. Since EVs inherit the cell recognition ability of the surface layer of the cell membrane from which they are derived, unique targeting is expected. In addition, EVs can fuse with the cell membrane and transport substances efficiently to the cytoplasm. Using EVs as carriers, it has become possible to deliver nucleic acid drugs, such as siRNA, protein drugs, and low-molecular-weight drugs.

### 6.2. Transfer and Preparation of Cargo by EVs

However, when using EVs as therapeutic DDS carriers, the vesicles need to be loaded with drugs to be delivered. There are two main methods for achieving this: pre-loading therapeutic nucleic acids, such as siRNA or miRNA, into the cells to be produced, and post-loading a drug externally into the produced EVs [137,138,139,140,141,142,143,144,145]. The former has drawbacks; for example, the mechanism by which contents such as nucleic acids are incorporated into EVs has not been fully elucidated. Although the method of physically encapsulating drugs in EVs does not have a high encapsulation rate, hydrophobic drugs, such as anti-cancer drugs, can be passively loaded through hydrophobic interactions with lipid bilayers. However, hydrophilic drugs, such as nucleic acids, require a technique to permeate through a hydrophobic lipid bilayer membrane. To date, some methods, such as electroporation and sonication, have been used to open pores by stimulating the EV membranes. However, in practice, the efficiency of loading drugs into EVs is extremely low. In addition, if excessive stimulation is applied, an aggregation of EVs may be induced, which may change the morphological characteristics and increase cytotoxicity by altering the surface potential of the membrane.

Since EVs have molecules with cell-recognition ability, such as membrane proteins, on their surface, they originally have a certain degree of targeting. In reality, however, when EVs are administered to the body, most of them accumulate in the liver and spleen, similarly to normal liposomes, and are finally cleared by macrophages in these organs [146]. Therefore, it is necessary to develop a method for efficient delivery to target tissues by enhancing the targeting ability of EVs. To date, approaches such as endogenous modification using biosynthetic processes, including genetic engineering and metabolic labelling, have been proposed [147,148,149]. However, endogenous modifications may affect cell function, and the properties of EVs may change from the original. When using EVs as therapeutic agents or DDS carriers, the exogenous EVs administered in vivo should be reliably delivered to the target cells. However, since a huge number of endogenous EVs are present in body fluids in vivo, it is difficult to obtain a therapeutic effect due to competition for cellular uptake, often resulting in insufficient therapeutic effects.

Even if EVs are derived from the same cell type or several populations with different biophysical properties, such as size, density, and morphology, and containing a variety of cargo elements, including proteins, lipids, nucleic acids, and glycans, it is still a major problem to define the efficacy of EVs for therapeutic drug application and the characteristics of the carrier when applied as a DDS carrier. However, to date, no optimal method has been established to fully understand this heterogeneity at the level of a single EV. Therefore, to realize the clinical application of EVs and evaluate reproducibility and safety from a pharmaceutical perspective, it is essential to develop a method for understanding complex and heterogeneous EVs. The clinical application of EVs requires the development of a cost-effective approach to industrially producing large amounts of EVs. However, most cells release EVs in very small amounts under normal conditions. Per minute, a single cell releases only EVs, cancer cells release several hundred, monocytes release less than a hundred, and mesenchymal stem cells produce less than one. Another bottleneck is the method of isolating EVs. The most commonly used method at present is ultracentrifugation, but this is associated with complicated operations and high equipment costs [15,150,151]. Various other methods with a range of advantages and disadvantages have been reported. Problems such as low recovery rates, small sample volumes that can be processed, and damage to EVs have been highlighted. Therefore, techniques for artificially modifying the functions of EVs are attracting attention for further medical applications. Regarding the genetic engineering of EVs, techniques such as modification of the EV membrane surface with functional membrane proteins and introduction of functional nucleic acids are currently the main methods in use. In addition, many methods of chemically and physically modifying the synthesized extracellular small molecules with functional synthetic molecules and macromolecules have been reported [152].

### 6.3. EVs-Based Vaccination

Additionally, dendritic cell-derived EVs express MHC (major histocompatibility complex) class I and II on the membrane surface and act as antigen-presenting cells. Thus, they facilitate the priming of cytotoxic T cells as well as antibody production. Thus, a novel cell-free vaccine therapy using cancer cell-derived EVs has been proposed because cancer cell-derived EVs contain cancer antigens. Administration of these EVs induces cancer antigen-specific immune response, and thereby provides an anti-tumor effect [153]. Although the possibility of inducing anti-tumor immunity by using EVs derived from cancer cells has been demonstrated, the effect was often inadequate. The reason for the issue is that the delivery efficiency of EVs to antigen-presenting cells and the subsequent antigen presentation efficacy are low, resulting in insufficient antigen presentation. Therefore, for the development of vaccine therapy using EVs derived from cancer cells, it is necessary to confer tropism to persistent antigen-delivering dendrite cells (DCs) by conferring retention of EVs at the administration site, and to control the intracellular dynamics of engulfed DCs. In addition, molecules contained in DC-derived EVs include molecules that trigger antigen presentation to CD^8+^ and CD^4+^ T cells and subsequent T cell proliferation, such as MHC I/II proteins and CD86. In addition, DC-derived EVs contain tetraspanins, including CD9, CD37, CD53, CD63, CD81, and CD82, which regulate DC interactions and are abundantly expressed. Thus, since DC-derived EVs contain many molecules involved in antigen presentation, they are expected to be applied as vaccine therapy. In fact, by inhaling virus-like particles (VLPs) in which EV is modified with the receptor binding domain (RBD) of the recombinant SARS-CoV-2 spike protein in mice or hamsters, the RBD was confirmed to be more tightly retained in both mucus-lined respiratory airways and lung parenchyma than liposome-based vaccines [154]. To date, mRNA vaccines have been approved against SARS-CoV-2 that are designed to induce systemic immunity via intramuscular injection. In addition, it is necessary to develop a cold chain at the actual place of vaccination. In contrast, EV-based vaccines can be stored at room temperature for three months after drying and target the lung specifically and effectively. Inhalation with an inhaler is also possible.

Additionally, there are several routes for viral budding, including direct budding from the cell membrane and budding via the MVB. Many enveloped RNA viruses, including retrovirus, flaviviruses, rhabdoviruses, and paramyxoviruses, interact with the ESCART complex and ESCART-associated proteins to facilitate budding from the plasma membrane [155]. In addition, hepatitis B virus (HBV) and hepatitis E virus (HEV) also interact with the ESCART complex and ESCART-associated proteins and are released via the MVB. Thus, viruses facilitate budding by hijacking the ESCRT machinery [156]. In addition to budding, viruses are known to transport viral genomes and virus-associated proteins to EVs by hijacking the ESCART mechanism, making EVs advantageous for virus survival. In fact, EVs derived from HIV (*human* immunodeficiency virus)-infected cells are known to contain Nef (negative regulatory factor), one of the HIV viral proteins, and cells that have taken up EVs containing Nef have increased susceptibility to HIV [157]. In addition, EVs derived from EBV-positive B-cell lymphomas are known to contain EBV-encoded microRNAs, and macrophages that have taken up these EVs induce changes in their properties similar to those of tumor-associated macrophages, promoting tumor cell proliferation [158].

### 6.4. Transfer and Preparation of Cargo by Liposome

Similar to EVs, liposomes are nanovesicles with a lipid bilayer membrane. Since there is a hydrophobic region between lipid bilayers and an internal aqueous phase, they can encapsulate both hydrophobic and hydrophilic drugs and reduce the toxicity of encapsulated drugs. Furthermore, techniques for preparing liposomes and loading functional molecules have been established. The composition of lipid membranes can be easily adjusted, and interior and surface modifications are relatively easy. By fusing such liposomes with EVs, it becomes possible to load drugs inside the EVs and modify the lipid composition and surface of the cell membrane [159,160]. One such technique for fusing EVs and liposomes involves using a fusogenic agent, polyethylene glycol (PEG), by which liposomes and EVs fuse through mixing so that more than 60% of membrane and soluble contents are translocated from liposomes to EVs under optimized conditions. By mixing these two types of particles and the gene plasmid, they can be efficiently loaded into the hybrid vesicle. A physical fusion method involves the use of an extruder, in which a thin lipid film used for liposome preparation is prepared, an EV dispersion is added to hydrate it, and hybrid vesicles with uniform sizes and high colloidal stability are then prepared by passing through 400 nm and 200 nm membrane filters using an extruder. Furthermore, by mixing hybrid vesicles and doxorubicin with an extruder, drug encapsulation is also possible. Moreover, liposomes and EVs are also fused using a physiochemical process called the freeze–thaw method, in which EVs and liposomes are mixed, frozen in liquid nitrogen, and thawed at room temperature to induce membrane fusion. This fusion proceeds relatively efficiently, regardless of the type of liposomal lipid or EVs. Furthermore, in hybrid vesicles incorporating various lipids, including neutral, anionic, cationic, and PEG-modified lipids, cellular uptake can be controlled by changing the type of lipid. In particular, hybrid vesicles containing PEG modifications have an increased uptake efficiency compared to unmodified EVs. With regard to constructing hybrid EVs using liposomes to incorporate viral fusogenic proteins that induce fusion under acidic conditions, it is possible to reconstitute functional membrane proteins into liposomes in one step by cell-free protein synthesis [161,162,163].

## 7. Conclusions

Various application methods are conceivable, such as protection from in vivo degradative enzymes by encapsulating nucleic acid drugs of interest in EVs and delivery systems that utilize the specificity of molecules present in the vesicle membrane. Thus, based on the possibility of nucleic acid therapy by DDS using EVs, nucleic acid medicine may become a novel drug delivery modality for COVID-19 vaccine development.

## Figures and Tables

**Figure 1 vaccines-11-00539-f001:**
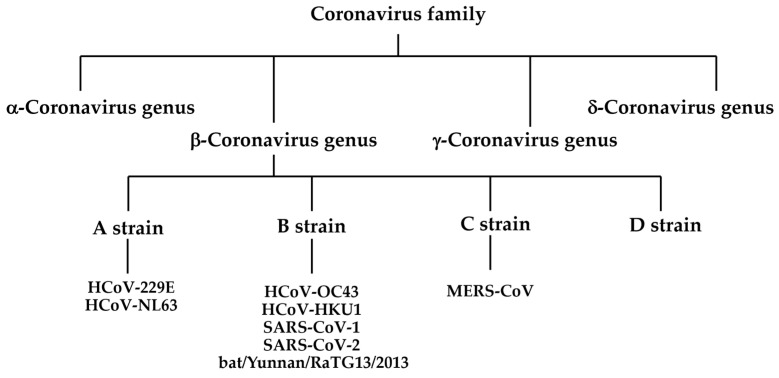
Classification of coronavirus family.

**Figure 2 vaccines-11-00539-f002:**
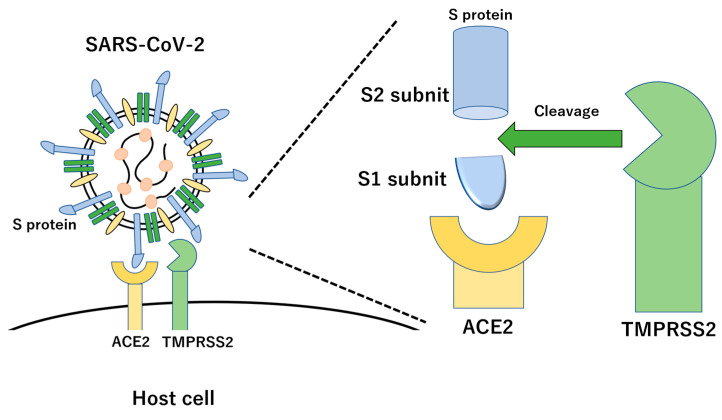
Molecular interaction of SARS-CoV-2 with host cell via S protein and ACE2 receptor. The S protein produces two subunits, the S1 and S2 subunits, with cleavage by TMPRSS2.

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
