# Peer review of "Extracellular Vesicle-Based SARS-CoV-2 Vaccine"

_vaccines, 2023, doi:10.3390/vaccines11030539_

Round 1

Reviewer 1 Report

Dear Authors,

The review on "Extracellular Vesicle-based SARS-CoV-2 Vaccine" is a relevant topic and interesting but more focus towards EVs will be interesting to the readers. There are some comments and suggestions.

1. Briefly explain about the biogenesis of the EVs

2. History of EV-based vaccines and their role in other viral diseases (table)

3. Briefly about function and application of COVID-19 EVs published recently

4.  Briefly about approved COVID-19 vaccines and why EV based vaccines will be advantageous.

Thanks

Author Response

The review on "Extracellular Vesicle-based SARS-CoV-2 Vaccine" is a relevant topic and interesting but more focus towards EVs will be interesting to the readers. There are some comments and suggestions.

  1. Briefly explain about the biogenesis of the EVs

Thank you very much for your review. According to reviewer’s comments, we corrected it by adding some sentence in page 9.

  1. History of EV-based vaccines and their role in other viral diseases (table)

Thank you very much for your review. According to reviewer’s comments, we corrected it by adding some sentence in page 11.

  1. Briefly about function and application of COVID-19 EVs published recently

Thank you very much for your review. According to reviewer’s comments, we corrected it by adding some sentence in page 11.

  1. Briefly about approved COVID-19 vaccines and why EV based vaccines will be advantageous.

Thank you very much for your review. According to reviewer’s comments, we corrected it by adding some sentence in page 11.

Reviewer 2 Report

The review elaborated by Matsuzaka and colleague represents a good overview of mRNA-based vaccines delivered by extracellular vesicles. However, I believe that this review needs to be supplemented with a brief but clear description of the different types of possible vaccines in addition to mRNA-based vaccines to further emphasize the importance of this type of formulation. This section could be added to "5. Vaccination of SARS-CoV-2". Finally, the section "6. Drug Delivery System by Extracellular Vesicle" should be divided into subsections to be more reader-friendly to highlight mechanisms such as uptake, pharmacokinetics, or pharmacodynamics of EVs, which are important in a vaccine formulation context itself. 

Author Response

The review elaborated by Matsuzaka and colleague represents a good overview of mRNA-based vaccines delivered by extracellular vesicles. However, I believe that this review needs to be supplemented with a brief but clear description of the different types of possible vaccines in addition to mRNA-based vaccines to further emphasize the importance of this type of formulation. This section could be added to "5. Vaccination of SARS-CoV-2". Finally, the section "6. Drug Delivery System by Extracellular Vesicle" should be divided into subsections to be more reader-friendly to highlight mechanisms such as uptake, pharmacokinetics, or pharmacodynamics of EVs, which are important in a vaccine formulation context itself. 

Thank you very much for your review. According to reviewer’s comments, we corrected it by adding some sentence in page 9 and sub-sections in section 6.

Round 2

Reviewer 1 Report

The Authors have made necessary changes.